# Redox-Regulated Pathways in Glioblastoma Stem-like Cells: Mechanistic Insights and Therapeutic Implications

**DOI:** 10.3390/brainsci15080884

**Published:** 2025-08-19

**Authors:** Nadia Fernanda Esteban-Román, Elisa Taddei, Edson Castro-Velázquez, Lorna Villafuentes-Vidal, Alejandra Velez-Herrera, Moisés Rubio-Osornio, Carmen Rubio

**Affiliations:** 1Faculty of Veterinary Medicine, Universidad Autónoma Metropolitana, Mexico City 04960, Mexico; nadia.esteban.roman@gmail.com; 2Department of Neurophysiology, Instituto Nacional de Neurología y Neurocirugía, Mexico City 14269, Mexico; e.taddei.l@gmail.com; 3Faculty of Medicine, University of Guadalajara, Guadalajara 44340, Mexico; edson.castro4440@alumnos.udg.mx; 4Mexican Faculty of Medicine, Universidad La Salle, Mexico City 14000, Mexico; lorna@gmail.com; 5University Center for Health Sciences, University of Guadalajara, Guadalajara 44340, Mexico; alejandra@gmail.com; 6Department of Neurochemistry, Instituto Nacional de Neurología y Neurocirugía, Mexico City 14269, Mexico; moises.rubio@innn.edu.mx

**Keywords:** glial stem-like cells, glioma, glioblastoma, oxidative stress, redox-targeted therapy, cancer stem cells

## Abstract

Glioblastoma (GBM) is the most aggressive primary brain tumor, characterized by rapid proliferation, invasiveness, therapeutic resistance, and an immunosuppressive tumor microenvironment. A subpopulation of glial stem-like cells (GSCs) within GBM tumors contributes significantly to tumor initiation, progression, and relapse, displaying remarkable adaptability to oxidative stress and metabolic reprogramming. Recent evidence implicates the atypical kinases RIOK1 and RIOK2 in promoting GBM growth and proliferation through their interaction with oncogenic pathways such as AKT and c-Myc. Concurrently, the redox-sensitive Nrf2/Keap1 axis regulates antioxidant defenses and supports GSC survival and chemoresistance. Additionally, aberrant activation of the canonical *Wnt/β*-catenin pathway in GSCs enhances their self-renewal, immune evasion, and resistance to standard therapies, particularly under oxidative stress conditions. This review integrates current knowledge on how redox homeostasis and key signaling pathways converge to sustain GSC maintenance and GBM malignancy. Finally, we discuss emerging redox-based therapeutic strategies designed to target GSC resilience, modulate the tumor immune microenvironment, and surmount treatment resistance.

## 1. Introduction

Brain cancer is one of the most aggressive and fatal forms of cancer. Scientists classify these tumors as primary or secondary based on their origin, either from cells within the central nervous system (CNS) or from metastatic cells that spread from other tissues through hematogenous dissemination [1]. Experts categorize primary tumors as glial or non-glial. Gliomas, a type of tumor that originates from glial cells, encompass astrocytomas, ependymomas, and oligodendrogliomas [2]. Non-glial tumors include meningiomas, medulloblastomas, and schwannomas, among others [3]. Gliomas constitute around 81% of malignant brain tumors and are generally linked to a poor prognosis. As of 2021, the World Health Organization (WHO) classifies gliomas into four categories according to histopathological and molecular features. Grade 1 gliomas are often well differentiated and display modest growth, while grades 2 and 3 demonstrate escalating levels of infiltration and anaplasia. Grade 4 gliomas, including IDH-wildtype glioblastoma (GBM), exhibit extreme aggressiveness and malignancy, resulting in the worst clinical outcomes [4]. In Mexico, glioblastomas (GBMS) constitute 28% of all gliomas and approximately 9% of all intracranial neoplasms [5]. Despite receiving appropriate care, this includes maximum surgical resection followed by concomitant/adjuvant radiotherapy and temozolomide, as described in the Stupp protocol [6]. Administration of drugs, such as rocarbazine, lomustine, and vincristine, reduces the most common symptoms [7]. Patients with GBM have an estimated mean survival of 12 to 16 months. This unfavorable prognosis is associated with the heterogeneity of the tumor, its capacity for microscopic infiltration of adjacent tissues, and the challenges in delineating tumor margins after surgical excision [8], presenting symptoms such as headache (50%), seizures (20–50%), neurocognitive impairment (30–40%), and focal neurological deficits (10–40%) [9]. A primary factor in glioma heterogeneity is the existence of glioma stem-like cells (GSCs). These cells have morphological and functional similarities to normal tissue stem cells and are distinguished by their exceptional resistance to cell death when exposed to radiotherapy and chemotherapy. This resistance leads to repeated treatment failure and tumor recurrence. GSCs are pivotal in sustaining tumor heterogeneity and are frequently located in perivascular and perinecrotic niches within the tumor microenvironment [9].

Oxidative stress (OS) plays a significant role in tumor control and progression. It affects gene expression, interferes with cell cycle regulation, and fosters unregulated tissue proliferation. Oxidative stress (OS) arises from a disparity between the production of reactive oxygen species (ROS) and the antioxidant defense systems that mitigate their effects. Reactive oxygen species (ROS), if not properly regulated, can facilitate rapid oncogenic transformation and tumor advancement [10]. Oxidative stress makes neural tissue particularly susceptible; thus, redox imbalance has been observed in numerous malignancies, including gliomas. Modified tumor metabolism causes this imbalance, which promotes circumstances that increase ROS levels. Thus, this oxidative microenvironment promotes the inactivation of tumor suppressor proteins. It drives adjacent healthy cells to generate antioxidants like catalase and superoxide dismutase (SOD) in a bid to reestablish redox equilibrium [11]. This review centers on glioma stem-like cells (GSCs) as pivotal contributors to the initiation, development, and recurrence of glioblastoma (GBM). It meticulously analyzes the processes by which redox imbalance and the activation of essential signaling pathways, namely Nrf2/Keap1, AKT/c-*Myc*, and *Wnt/β*-catenin, facilitate the survival, plasticity, and significant treatment resistance of GSCs, as well as the capacity of these cancer stem-like cells to adapt to oxidative stress and modify their metabolism, hence facilitating tumor proliferation and resistance to therapy. The paper examines the intricate interactions among oxidative stress, metabolic reprogramming, and the immunosuppressive tumor microenvironment, emphasizing their combined influence on GSC-driven tumor aggressiveness and recurrence. Ultimately, prospective redox-oriented therapy options especially aimed at targeting GSCs are examined, with the objective of restoring redox equilibrium, surmounting treatment resistance, and enhancing existing GBM therapies.

## 2. Materials and Methods

We conducted an extensive literature review utilizing PubMed, Scopus, and Google Scholar to uncover pertinent research investigating the molecular underpinnings of glioblastoma (GBM) growth, emphasizing glial stem-like cells (GSCs), oxidative stress, and critical signaling pathways. We included peer-reviewed articles published in English from the last 15 years, prioritizing studies that addressed molecular mechanisms, redox signaling, GSC biology, and therapeutic strategies. We excluded reports lacking experimental support or relevance to glioma stem-like cells, as well as non-original articles without critical insight. The search method utilized several keyword combinations, encompassing glioblastoma, glial stem-like cells, GSCs, oxidative stress, redox homeostasis, Nrf2/Keap1, *Wnt*/β-catenin, RIOK1, RIOK2, AKT, c-Myc, immunological microenvironment, metabolic reprogramming, chemoresistance, and targeted therapeutics. We incorporated experimental research and review articles that utilized validated GBM models to examine the role of GSCs and related signaling pathways in tumor proliferation, invasion, treatment resistance, and immune evasion. We chose studies based on their methodological rigor, scientific significance, and contribution to elucidating the molecular and cellular mechanisms underlying GBM development and treatment resistance.

Analyzed studies identified common and specific molecular mechanisms by which redox balance, metabolic adaptation, and key signaling networks (Nrf2/Keap1, *Wnt*/β-catenin, RIOK1/2, AKT, c-Myc) govern GSC maintenance, tumor progression, immune suppression, and therapeutic response. We placed special emphasis on research clarifying the interaction between oxidative stress responses and oncogenic pathways that enhance GSC survival and plasticity. We combined the findings to offer a comprehensive perspective on the intersection of redox biology, stemness, and oncogenic signaling in GBM, emphasizing innovative therapeutic methods designed to disrupt GSC-mediated resistance and alter the immunosuppressive tumor microenvironment.

## 3. The Function of Human Aldehyde Dehydrogenase in Redox Equilibrium in Oncology

The human aldehyde dehydrogenase (ALDH) enzyme family is responsible for the oxidation of aldehydes, consequently safeguarding cells against oxidative stress. These enzymes are situated in many cellular compartments such as the cytoplasm, mitochondria, nucleus, and endoplasmic reticulum, where they carry out detoxification functions [10,12,13,14]. ALDH enzymes mitigate oxidative stress in stem cells, particularly GSCs, which are implicated in tumor advancement and resistance to therapy [15,16,17,18]. The mitochondrial enzyme aldehyde dehydrogenase 1 family member L2 (ALDH1L2) is crucial for sustaining redox equilibrium via the synthesis of reduced nicotinamide adenine dinucleotide phosphate (NADPH). ALDH1L2 facilitates the transformation of 10-formyl-tetrahydrofolate (10-formyl-THF) into carbon dioxide (CO_2_) and tetrahydrofolate (THF), while producing NADPH. Figure 1 illustrates how NADPH acts as a cofactor for glutathione reductase, facilitating the reduction of oxidized glutathione (GSSG) to its reduced form (GSH). Reduced glutathione subsequently works as a cofactor for glutathione peroxidase (GPX), an enzyme that enhances reactive oxygen species (ROS) and preserves cellular redox equilibrium. Experimental evidence in GSCs suggests that ALDH1L2 bolsters the antioxidant defenses crucial for stem cell survival, self-renewal, and resistance to therapy-induced oxidative stress. The depletion of ALDH1L2 compromises NADPH synthesis, elevates intracellular reactive oxygen species (ROS), diminishes neurosphere formation, and reduces the expression of stemness markers, underscoring its significance in maintaining the aggressive phenotype of GSCs [19,20,21,22,23,24,25,26]. Moreover, ALDH activity functions as a biomarker for the identification and isolation of cancer stem cells, particularly GSCs associated with heightened therapeutic resistance and tumor recurrence [27,28,29,30]. Therefore, it is essential to regulate the oxidative stress mechanisms that influence glioblastoma substances.

## 4. The HA-CD44 Axis Regulates the Oxidative Stress Response in Glioblastoma

Hyaluronic acid (HA) is a glycosaminoglycan and an essential constituent of the extracellular matrix in the nervous system, which interacts with its receptor CD44. This contact is essential for brain development, facilitating the proliferation and maturity of neural stem cells, along with cellular repair. Under healthy settings, HA associates with the extracellular “N” domain of CD44. Subsequently, metalloproteinases and secretases proteolytically cleave this domain, which results in the release of the intracellular domain CD44-IC. CD44-ICD translocates to the nucleus to modulate numerous pathways, including the activation of ezrin/radixin/moesin (ERM) proteins, the tumor suppressor protein merlin, and transcription factors such as RUNX2 [31,32,33,34,35,36,37,38,39,40]. The HA-CD44 axis in GBM is involved in the regulation of epithelial–mesenchymal transition (EMT), cellular proliferation, invasion, and metastasis. CD44, particularly its variant isoform CD44v, functions as a biomarker for GSCs [32,33,34,41,42,43]. The binding of HA to CD44 activates many receptor tyrosine kinase (RTK) pathways, including epidermal growth factor receptor (EGFR) and ErbB2, as well as non-receptor Src family kinases. These activations initiate downstream signaling cascades that encompass Ras and RhoGTPase, mitogen-activated protein kinases (MAPK), phosphoinositide 3-kinase (PI3K), and signal transducer and activator of transcription 3 (STAT3), thereby facilitating tumor cell survival and proliferation [22,23,24,25,26,27,28,29,30,31,32,33,34,41,42,43]. An elevated CD44 expression is associated with a worse prognosis and conveys therapeutic resistance through oxidative stress responses. The reduction of CD44 enhances the susceptibility of glioma cells, particularly GSCs, to oxidative stress triggered by chemotherapeutic drugs such as temozolomide (TMZ) and carmustine (BCNU) [44,45]. ROS activates NF2 tumor suppressor protein merlin, consequently initiating the *Hippo* tumor suppressor pathway. This pathway entails the phosphorylation and activation of Mammalian Sterile Twenty-like kinases 1 and 2 (MST1/2) and large tumor suppressor homologs 1 and 2 (LATS1/2), resulting in the phosphorylation and inactivation of the Yes-associated protein (YAP). Inactivation of YAP diminishes the production of anti-apoptotic proteins cIAP1/2, elevates cleaved caspase-3 levels, and facilitates apoptosis in glioma cells [33,34,42,45,46,47,48,49]. Furthermore, Figure 2 shows how the presence of ROS interferes with the interaction between HA and its receptor CD44 by oxidizing the structure of HA [50]. This obstructs ligand binding by the receptor, thereby decreasing signaling pathways associated with cell proliferation, survival, and invasion. Besides its involvement in tumor growth, researchers acknowledge CD44 as a principal biomarker of cancer stem cells (CSCs). These cells are associated with tumor initiation, therapeutic resistance, and recurrence in glioblastoma. Studies have demonstrated that reactive oxygen species (ROS) regulate the stemness and viability of CSCs by affecting redox-sensitive signaling pathways, including those associated with CD44, Notch, and *Wn*t/β-catenin [51,52]. Increased ROS levels can cause oxidative alterations that disrupt CD44-HA interactions, hence diminishing CSCs’ viability and self-renewal potential [53]. These data endorse the concept that targeting CD44, in conjunction with the manipulation of redox equilibrium, constitutes a viable therapeutic approach for glioblastoma. This redox imbalance has a significant impact on various genetic and metabolic pathways, reinforcing the need to further explore the molecular mechanisms that allow these cells to adapt. Therefore, it is crucial to analyze the function of key proteins like the mitochondrial chaperone TRAP1, whose activity is directly related to the regulation of mitochondrial metabolism and ROS generation, positioning it as a central regulator in glioblastoma adaptation and survival.

## 5. Genetics and Oxidative Pathways in Glioblastoma Multiforme

Tumor necrosis factor receptor-associated protein 1 (TRAP1), or heat shock protein 75 (HSP75), is a mitochondrial chaperone belonging to the HSP90 family, encoded by the TRAP1 gene on chromosome 16p13. It has a crucial role in regulating mitochondrial metabolism, primarily by inhibiting succinate dehydrogenase (SDH), hence diminishing the generation of reactive oxygen species (ROS). Furthermore, TRAP1 performs antioxidant and anti-apoptotic roles in tumor cells via ERK1/2-dependent activation [54]. It stabilizes hypoxia-inducible factor 1α (HIF1α), which facilitates a metabolic transition towards aerobic glycolysis while inhibiting oxidative phosphorylation (OXPHOS). This enhances the Warburg effect and the activation of genes that promote angiogenesis, contributing to cancer progression.

Conversely, its association with mitochondrial tyrosine–protein kinase c-Src inhibits cytochrome c oxidase activity, affecting its capacity to promote oxidative phosphorylation. Researchers have identified an opposing role in GBM. Figure 3 illustrates the mechanism by which TRAP1TRAP1 stabilizes and activates SDH, enhancing mitochondrial respiration [55]. An experimental study confirmed that inhibition of TRAP1 led to reduced mitochondrial respiration, increased lactate production, and elevated ROS generation, which were attributed to a decrease in SIRT3 expression directly proportional to TRAP1 levels [56]. Studies primarily document elevated TRAP1 expression in grade IV brain tumors; thus, this expression is associated with a worse prognosis, with even more TRAP1 overexpression in GSC, unlike in normal astrocytes and other GBM-derived cell lines, indicating an adaptation mechanism that promotes metabolism reliance on more efficient mitochondrial respiration [56]. These findings align with growing evidence that GSCs, a subset of CSCs in GBM, demonstrate increased mitochondrial adaptability and ROS scavenging, thereby facilitating their survival and therapeutic resistance [57,58]. Additionally, TRAP1 engages with cyclophilin D (CypD), obstructing the opening of the mitochondrial permeability transition pore (mPTP) and inhibiting apoptosis in neoplastic cells [54]. While we incompletely understand its role in healthy cells, numerous studies indicate its significant participation in the pathophysiology of various malignancies, including GBM [54,55,56,57]. Key proteins such as TRAP1 and SIRT3 regulate mitochondrial metabolism and antioxidant defenses. TRAP1 inhibits apoptosis, while SIRT3 activates enzymes like manganese superoxide dismutase (MnSOD) to reduce ROS. The interaction between TRAP1 and SIRT3 is fundamental for maintaining mitochondrial homeostasis and enabling GSCs to thrive under stress.

## 6. SIRT3: Mitochondrial Regulator of Redox Equilibrium and Cellular Oxidative Stress

Sirtuin 3 (SIRT3), a protein situated at locus 11p15.5 on chromosome 11, encodes a NAD^+^-dependent class III histone deacetylase that is located in the mitochondria. This enzyme is crucial in the response to oxidative stress by facilitating the deacetylation and activation of essential antioxidant enzymes, including manganese superoxide dismutase (MnSOD) and isocitrate dehydrogenase (IDH). This method facilitates the reduction of ROS production to levels conducive to cellular viability and proliferation, therefore enhancing cellular resilience to oxidative stress, maintaining mitochondrial membrane integrity, and promoting mitochondrial homeostasis [59,60].

SIRT3 has been demonstrated to induce instability and inactivation of hypoxia-inducible factor 1 alpha (HIF-1α) [61,62], disrupting signaling pathways related to cellular adaptation to hypoxic environments, as illustrated in Figure 4 [59]. Recent studies have emphasized the significance of SIRT3 in GSCs, where it regulates redox equilibrium and promotes stemness and survival under conditions of oxidative stress [60]. Lysine acetylation significantly influences the metabolic phenotype of glioblastoma cells, and SIRT3 is essential for regulating the balance of metabolic processes in these cancer cells [60,62].

SIRT3 knockdown in GSCs disrupts mitochondrial respiration and increases ROS buildup, resulting in diminished tumorigenicity [60]. SIRT3 modulates the antioxidant defense system in GSCs, improving their resistance to radiation by preserving redox homeostasis. These findings emphasize the significance of SIRT3 as a metabolic and redox regulator in glioblastoma stem cells and reinforce the call for further research into its potential as a therapeutic target within this particular subgroup of tumor cells. SIRTs have been shown to play important regulatory roles in almost all cellular functions, including mitochondrial biogenesis, oxidative stress, inflammation, cell growth, energy metabolism, neural function, and stress resistance [62,63]. Preclinical studies have identified several sirtuin modulators—both inhibitors and activators—that alter tumor growth, sensitize cells to temozolomide, and regulate pathways such as JAK2/STAT3, NF-κB, and mitochondrial metabolism [63,64].

## 7. Functional Interdependence of TRAP1 and SIRT3 in the Metabolic Adaptation of Glial Stem Cells in Glioblastoma

There exists a bidirectional functional relationship between TRAP1 and SIRT3 [65]. TRAP1 enhances the enzymatic activity of SIRT3, leading to reduced mitochondrial ROS generation, facilitated by the deacetylation and activation of SOD2. Simultaneously, SIRT3 enhances TRAP1 acetylation, indicating a positive feedback loop between both proteins [66]. The TRAP1-SIRT3 functional axis enhances mitochondrial respiration efficiency by quantitatively and qualitatively regulating the electron transport chain (ETC) in GSCs, especially under hypoglycemic circumstances. The mechanisms are illustrated in Figure 4 [67]. These findings correspond with previous studies, which demonstrated that GSCs have increased metabolic plasticity, enabling them to adapt to nutrient-deficient microenvironments by prioritizing oxidative phosphorylation [68]. Mitochondrial control, encompassing the activity of proteins such as TRAP1 and SIRT3, is crucial for sustaining redox homeostasis and stemness in CSCs [58,63]. The targeted suppression of either TRAP1 or SIRT3 correlates with a reduction in the other protein, indicating a significant functional dependency. These findings reinforce the involvement of these proteins in the carcinogenesis of GSCs in GBM by enabling an adaptive metabolic shift that enhances cellular self-renewal and survival under conditions of nutritional deprivation and tumor microenvironment stress [55,58]. The regulation of redox equilibrium in GSCs also involves other key mitochondrial players, such as the sirtuin SIRT3. This enzyme, crucial for the oxidative stress response, functions as a histone deacetylase that activates other antioxidant enzymes, such as manganese superoxide dismutase (Mn-SOD), thereby reducing ROS production and strengthening cellular resilience. Therefore, the interaction of proteins like TRAP1 and SIRT3 is fundamental for maintaining mitochondrial homeostasis and the ability of GSCs to thrive under stressful conditions.

## 8. Modifications in the PI3K/AKT/mTOR Pathway in Glioblastoma Molecular and Prognostic Significance

The PI3K/AKT/mTOR pathway is a major signal transduction route that regulates cell growth, proliferation, metabolism, and survival. Researchers categorize phosphoinositide-3-kinases (PI3Ks), which are lipid kinases associated with the plasma membrane, into three classes: IA, IB, II, and III. Class IA, significant in oncogenesis, consists of a regulatory subunit (p85), encoded by the PIK3R1 (5q13.1), PIK3R2 (19p13.11), and PIK3R3 (1p34.1) genes, and a catalytic subunit (p110), encoded by PIK3CA (3q26.32), PIK3CB (3q22.3), and PIK3CD (1p36.22) [26,27,28,29,30,31]. Conversely, AKT, or protein kinase B (PKB), a serine/threonine kinase, serves as the primary downstream effector. This protein comprises three isoforms that the AKT1 (14q32.33), AKT2 (19q13.2), and AKT3 (1q43-q44) genes encode. Various extracellular stimuli uniquely activate these isoforms [69,70,71] (Figure 5). The initiation of this pathway occurs when growth factors, including EGFR, fibroblast growth factor (FGFR), insulin-like growth factor type I (IGF-IR), cytokines, and hormones bind to receptor tyrosine kinases (RTKs), triggering their autophosphorylation. This facilitates the activation of PI3K class IA, whose catalytic subunit transforms PIP2 into PIP3.

PIP3 functions as a second messenger, recruiting AKT to the plasma membrane, where it undergoes phosphorylation and activation. AKT phosphorylates various substrates, including glycogen synthase kinase 3 (GSK3), thereby modifying cellular metabolism; members of the FOXO transcription factor family (FOXO1, FOXO3A, FOXO4), which suppresses their antiproliferative effects; and tuberous sclerosis complex 2 (TSC2), diminishing its inhibitory role on the Ras homolog enriched in brain (RHEB), thus promoting mTORC1 activation. The latter facilitates cellular proliferation, angiogenesis, protein synthesis, and energy storage, with activity dynamically modulated in accordance with intracellular energy levels [72,73]. The PI3K/AKT/mTOR pathway is typically hyperactivated in neoplastic cells, promoting tumor survival and proliferation. Somatic mutations in PIK3CA and PIK3R1 in GBM, correlating with a poor prognosis [74]. Research has proven that GSCs depend significantly on the PI3K/AKT/mTOR pathway to maintain their self-renewal and therapeutic resistance [75]. Past research has demonstrated that AKT activation facilitates the preservation of GSCs by endorsing transcriptional pathways associated with stemness. Rascio, F., et al. (2021) determined that the suppression of PI3K hinders the proliferation of GSCs and their ability to produce neurospheres [72]. Phosphatase and Tensin Homolog (PTEN) (10q23.3), a major negative regulator of this pathway, functions as a tumor suppressor lipid phosphatase that dephosphorylates PIP3, thus inhibiting AKT recruitment and activation. The loss or mutation of PTEN results in the persistent activation of the PI3K/AKT/mTOR pathway and is linked to malignant processes [72,73,75,76]. In addition, the absence of PTEN is associated with the proliferation of GSC populations and increased resistance to radiotherapy, accentuating the clinical significance of targeting this pathway in GBM [77,78]. In a study by Praisthy et al. (2025), cytotoxicity was evaluated in human SH-SY5Y neuroblastoma cells at 24, 48, and 72 h. In vitro assays showed a dose-dependent increase in superoxide dismutase (SOD) (from 3.18 to 18.42 U/mg protein) and reduced glutathione (GSH) (from 7.79 to 22.03 µmol/g tissue) [79]. The absence of the PTEN protein leads to sustained activation of the PI3K/AKT/mTOR pathway, which is associated with glioma stem cell (GSC) proliferation and resistance to radiotherapy. This activation also influences the cells’ ability to manage oxidative stress, as shown by increased antioxidant levels (SOD and GSH) in studies. Therefore, the crucial role of the PI3K/AKT/mTOR pathway in GSC survival, proliferation, and resistance makes it a highly attractive therapeutic target for combating glioblastoma aggressiveness and recurrence.

## 9. Targeting the PI3K/AKT/mTOR Pathway in Glioblastoma Stem Cells for Therapeutic Intervention

The distinctive function of the PI3K/AKT/mTOR pathway in glioblastoma stem cells (GSCs) has attracted heightened interest owing to its relevance in tumor sustenance, resistance, and recurrence. GSCs demonstrate increased activation of this pathway, which facilitates self-renewal, proliferation, metabolic adaptability, and resistance to apoptosis, particularly in hypoxic and nutrient-deficient environments. It is known that PI3K/AKT signaling is crucial for the stem-like characteristics and viability of GSCs in the perivascular environment. Likewise, pharmacological inhibition of mTOR interferes with GSC proliferation, neurosphere formation, and increases cell sensitivity to temozolomide [80]. Incidentally, researchers have observed differential expressions of AKT isoforms in GSCs in relation to differentiated tumor cells, indicating isoform-specific activities. AKT3 is notably correlated with the preservation of stemness markers and is associated with increased tumor aggressiveness in GSCs [81]. These findings substantiate the concept that inhibiting the PI3K/AKT/mTOR pathway in GSCs not only impedes tumor proliferation but may also avert recurrence by eliminating therapy-resistant subpopulations (Table 1).

Inhibiting the PI3K/AKT/mTOR pathway is a promising strategy against glioblastoma, as AKT3 expression is associated with GSC aggressiveness. However, this strategy may be more effective when combined with the modulation of the Nrf2/Keap1 pathway. Nrf2/Keap1 is crucial for redox balance and protecting cells from oxidative stress, making it a key factor for tumor survival. This suggests that targeting both pathways simultaneously could be more effective in combating tumor resistance and recurrence.

## 10. The Nrf2/Keap1 Pathway and Its Significance in Oxidative Stress and Tumor Proliferation

The Nrf2/Keap1 signaling pathway is essential for sustaining cellular redox homeostasis and for the regulation of tumor development via the modulation of oxidative stress responses. Nuclear factor erythroid 2-related factor 2 (Nrf2) is a transcription factor belonging to the Cap ‘n’ Collar (CNC) family, encoded by the NFE2L2 gene located at 2q31.2. It comprises seven functional domains (Neh1-Neh7) crucial for the activation of antioxidant response element (ARE)-regulated genes pertinent to redox regulation, β-globin production, and detoxification mechanisms [82,83]. The KEAP1 gene, located on 19p13.2, encodes Keap1 (Kelch-like ECH-associated protein 1) and is a cytoplasmic protein that negatively modulates Nrf2 through its five domains (three BTB, one IVR, and two DGR domains), promoting Nrf2 ubiquitination and subsequent proteasomal degradation via the Cul3-Roc1 E3 ligase complex under basal conditions [83,84]. The interaction between the DLG and ETGE motifs in the Neh2 domain of Nrf2 and the DGR domain of Keap1 facilitates this regulation; conversely, Nrf2 degradation may also occur independently of Keap1 through the Neh6 domain and β-TrCP-mediated ubiquitination [83].

Oxidative stress leads to the alteration of essential cysteine residues on Keap1, including Cys151 in the BTB domain and Cys273 and Cys288 in the IVR domain, which disturbs its interaction with Cul3 and Nrf2, inhibiting ubiquitination and facilitating Nrf2 stability and nuclear translocation. Nrf2 heterodimerizes with sMaf proteins in the nucleus to bind to antioxidant response elements (AREs), thereby promoting the transcription of antioxidant enzymes such as NAD(P)H quinone oxidoreductase 1 (NQO1), heme oxygenase-1 (HO-1), aldo-ketoreductase 1C1 (AKR1C1), and thioredoxin (Trx), along with genes involved in glutathione (GSH) synthesis and recycling, including γ-glutamylcysteinyl ligase (GCL), the cystine/glutamate transporter xCT, glutathione reductase (GR), and GSH synthetase [85]. The Neh3, Neh4, and Neh5 domains facilitate transcription by attracting coactivators, including CHD6, CBP/p300, and RAC3/SRC-3, but Neh7 inhibits Nrf2 via association with retinoid X receptor α (RXRα) [86]. Recent research emphasizes the critical function of Nrf2 in CSCs, especially in GBM. Figure 6 illustrates how the overexpression of Nrf2 in GBM associates with an elevated histopathological grade and contributes to preserving the redox equilibrium essential for the survival and proliferation of CSCs [87]. A previous study demonstrated that the knockdown of Nrf2 in U251 glioblastoma cells resulted in diminished expression of target genes, including GCLC and GR, leading to a 70% reduction in GSH levels and a 117.4% increase in ROS. Researchers quantified this impairment in proliferation as a 37.1% and 45.2% decrease on days 3 and 4, respectively, along with the inhibition of the AKT and ERK1/2 pathways [88,89]. Interestingly, the addition of monoethyl glutathione (GMEE) to cells reinstated GSH levels and AKT signaling, but it did not restore ERK1/2 activity, highlighting the essential role of intracellular GSH in Nrf2-mediated proliferation, independent of ROS scavenging. Accordingly, recent research in stem cell biology has demonstrated that Nrf2 activity is crucial for preserving the self-renewal and differentiation capabilities of many stem cell types by safeguarding them against oxidative damage. Nrf2 promotes the survival of neural stem cells during oxidative stress by upregulating antioxidant defenses, facilitating neurogenesis and tissue regeneration [89]. However, in CSCs, Nrf2 activation promotes chemoresistance and enables metabolic reprogramming that facilitates tumorigenicity [81]. Additionally, Nrf2 augments radioresistance and upholds stemness in GSCs via the regulation of redox homeostasis and mitochondrial function [90]. These findings establish Nrf2 as a dual-function entity in tumor biology: it safeguards normal stem cells from oxidative damage, yet its aberrant activation in CSCs promotes tumor progression and therapeutic resistance. The Nrf2 transcription factor has a dual role in glioblastoma: while it protects normal cells from oxidative stress, its activation in cancer stem cells (CSCs and GSCs) promotes resistance to chemotherapy and radiotherapy, as well as tumor survival. These adaptive mechanisms in GSCs do not act in isolation; they connect with other key pathways. One such pathway is *Wnt/β*-catenin signaling, which is essential for nervous system development and, in cancer, contributes to GSC proliferation and aggressiveness. Therefore, cancer benefits from the integration between the oxidative stress response and developmental pathways.

## 11. Impact of Oxidative Stress on the Wnt/β-Catenin Signaling Pathway in Neoplastic Cells

The *Wnt*/β-catenin signaling pathway is essential for the embryonic development of the nervous system. The canonical pathway initiates with the interaction of WNT ligands with the membrane receptors Frizzled (FZD) and LRP5/6. The interaction facilitated by Dishevelled (DVL) proteins results in the deactivation of the *β*-catenin degradation complex, consisting of adenomatous polyposis coli (APC), glycogen synthase kinase-3β (GSK-3β), AXIN scaffold protein, and casein kinase 1 (CK-1). This inactivation permits β-catenin to migrate to the nucleus, where it associates with T-cell factor/lymphoid enhancer-binding factor (TCF/LEF) and initiates the transcription of target genes, including c-myc, cyclin D, and VEGF [87,88,89]. In contrast, two primary pathways can initiate non-canonical *Wnt* signaling. The initial process entails alterations in cell polarity facilitated by DVL activation, which then activates the small GTPases Rho and Rac, resulting in the activation of c-Jun N-terminal kinase (JNK). The second modulates intracellular signaling through calcium mobilization via DVL-activated phospholipase C (PLC), resulting in the production of inositol 1,4,5-trisphosphate (IP3), which induces calcium release from the endoplasmic reticulum and activates signaling proteins, including PKC, Cdc42, CAMKII, TAK1, and NFAT [91,92,93,94]. In tumor cells, especially GBM, the *Wnt* pathway plays a key role by regulating GSC proliferation, migration, and inflammation. The *Wn*t signaling pathway facilitates the sustenance and differentiation of GSCs, consequently enhancing tumor heterogeneity and resistance to therapy. Aberrant activation of the *Wnt* pathway, often due to mutations in APC, AXIN, β-catenin, or TCF4, leads to prolonged self-renewal and stemness of GSCs [93,94,95,96,97,98]. As Figure 7 illustrates, oxidative stress can influence *Wnt/*β-catenin signaling via multiple routes. It upregulates DVL activation, thereby enhancing β-catenin stability and nuclear translocation, which facilitates tumor proliferation and invasion. Furthermore, ROS induce the production of c-*myc*, STAT, and the PI3K/Akt signaling pathways in cancer cells. Increased c-*myc* can promote FOXO transcription factors, raising ROS levels and triggering autophagy, especially in GSCs [95,99,100]. Hoogeboom et al. established that oxidative stress amplifies the interaction between β-catenin and FOXOs, which competes with TCF binding, influencing cell cycle arrest and regulating the balance between proliferation and quiescence [101].

NADPH oxidase 1 (Nox1) acts as a promoter of ROS within *Wnt/β*-catenin signaling. Nucleoredoxin (NRX), a redox-sensitive inhibitor, interacts with DVL to limit β-catenin activation. The oxidation and inactivation of NRX by ROS that Nox1 produces alleviates this inhibition, augmenting β-catenin activity and facilitating tumor growth through the disruption of redox homeostasis [95,102,103,104,105]. Ultimately, *Wnt/β*-catenin regulates metabolic reprogramming. It activates the Akt-mTOR pathway, facilitating aerobic glycolysis and enhancing glucose-6-phosphate dehydrogenase expression. In GSCs, mTORC1 activation induced by ROS promotes autophagy as an adaptation mechanism to redox stress, enforcing cell survival and tumor proliferation under unfavorable conditions [95,99,106,107]. The Wnt/β-catenin pathway and oxidative stress are connected in glioblastoma: the Nox1 protein generates ROS, which amplifies β-catenin activity and drives tumor growth. H_2_O_2_ also activates the Akt-mTOR pathway, promoting autophagy and glutathione transport for cell survival. The glioblastoma aggressiveness is the result of the interplay between these signaling pathways and the responses to oxidative stress [107,108].

## 12. Oxidative Stress Induced by Hydrogen Peroxide

The breakdown of oxidized or oxidatively altered proteins is a vital component of the cellular antioxidant defense mechanism. A prominent example is 4-hydroxy-2-nonenal (4-HNE), a principal reactive aldehyde that lipid peroxidation generates, which causes considerable cellular damage [108,109]. Hydrogen peroxide (H_2_O_2_) is considered the most suitable for redox signaling and is a potential candidate as a key molecule that determines the fate of cancer cells [110]. Research involving human leukemic U937 cells demonstrated that cathepsin G degrades 4-HNE-modified glyceraldehyde-3-phosphate dehydrogenase (GAPDH) [111,112]. In U373 human glioma cells, hydrogen peroxide (H_2_O_2_) treatment reduced GAPDH activity and protein levels in a dose-dependent manner [111,113]. In GSCs, H_2_O_2_ triggers the rapid transport of endogenously expressed glutathione to the cell surface. This leads to a threefold increase in cystine uptake following H_2_O_2_ exposure [114]. Furthermore, the kinetics of the increased cystine transport and cell surface expression have been shown to coincide with the kinetics of intracellular glutathione recovery after acute hydrogen peroxide exposure [115]. The H_2_O_2_ treatment diminished the activity of proteasomal enzymes (peptidylglutamyl-peptide hydrolase and chymotrypsin-like), as well as cathepsin G, as Figure 8 illustrates. Nonetheless, trypsin-like proteasomal activity escalated and was associated with the degradation of oxidatively modified GAPDH, a process obstructed by the proteasome inhibitor lactacystin [113,116]. The data suggest that proteasomal degradation pathways are crucial for the removal of damaged proteins during oxidative stress [111,116,117]. Hydrogen peroxide (H_2_O_2_) helps tumors survive by affecting protein degradation. While H_2_O_2_ damages some enzymes, it also activates others to remove damaged proteins like GAPDH. This cellular response extends to the tumor microenvironment, where oxidative stress activates the NR4A2 protein in microglia, which, in turn, promotes tumor growth. In short, the cellular response to oxidative stress is not an isolated event; it has a domino effect that increases glioblastoma aggressiveness.

## 13. Therapeutic Approaches for Tumor Immune Microenvironment

Single-cell sequencing of the GBM tumor immunological microenvironment has demonstrated that microglia undergo considerable oxidative damage. This stress triggers the activation of nuclear receptor subfamily 4 group A member 2 (NR4A2), facilitating transcriptional alterations that enhance tumor proliferation [117,118,119]. The targeted deletion of NR4A2 in myeloid cells (c) or its heterozygous knockout (Nr4a2^+/–^) has been shown to alter microglial plasticity in vivo, diminishing the population of variably activated microglia while augmenting their antigen-presenting capability to CD8+ T cells [120,121]. Activation of NR4A2 in microglia results in elevated expression of squalene monooxygenase (SQLE), which disrupts cholesterol homeostasis and promotes ROS production. This oxidative imbalance activates signaling pathways, including c-Myc, which contributes to the regulation of autophagy and tumor development [122,123]. However, the development of new therapeutic methods is hindered because the blood–brain barrier (BBB) restricts drug penetration. The transferrin receptor (TfR) is highly expressed by brain endothelial cells and GBM cells, and it is considered a promising target for GBM drug delivery [124].

## 14. Depletion of Glutathione

The tumor microenvironment (TME) significantly contrasts with normal tissue, presenting an acidic pH, hypoxia, ROS levels, and an enhanced antioxidant system [113]. Excessive ROS levels induce extensive damage to DNA, proteins, and lipids, ultimately resulting in cell death [125,126]. The antioxidant defense system comprises enzymatic elements such as superoxide dismutase (SOD), catalase (CAT), glutathione peroxidase (GPX), and thioredoxin (Trx). It also comprises non-enzymatic compounds like glutathione (GSH), ascorbic acid, and tocopherol [127]. Targeting the GSH system has emerged as a redox-sensitive strategy in pharmacologic intervention. Drug delivery techniques that harness the elevated intracellular GSH gradient in tumor cells, such as disulfide bonds, facilitate redox-responsive release of chemotherapeutic drugs [128]. These methods have demonstrated considerable effectiveness in GSCs, wherein increased antioxidant capacity increases resistance, as Figure 9 illustrates [129]. Previous research has demonstrated that the particle size of nanocarriers diminished following redox activation, therefore improving drug bioavailability and cellular absorption [130,131,132].

## 15. Inhibition of the EGFR/AKT Pathway

Metabolic reprogramming is central to the progression and recurrence of GBM, as it fulfills the energy requirements of rapidly proliferating tumor cells. The amplification of the EGFR and an elevated incidence of EGFRvIII mutations in GBM, and they associate these findings with aggressive tumor characteristics. In clinical samples, single-cell RNA sequencing and untargeted metabolomics have shown that the inhibition of receptor tyrosine kinase (RTK) and mevalonate pathways via MK-2206 and MK-803 impairs energy metabolism in EGFR-activated GBM [133,134]. Tumors exhibiting elevated EGFR expression presented increased lipid remodeling and cholesterol retention. Inhibition of the EGFR/AKT pathway diminished the tricarboxylic acid (TCA) cycle and ATP synthesis, impacting essential metabolic regulators [135,136]. This suppression also curtailed NF-κβ-dependent production of metabolic enzymes, including ACSS3, ACSL3, and ELOVL2.

The suppression of the mevalonate pathway decreased EGFR membrane concentrations, hindering subsequent signal transduction. The effects amplified the cytotoxic efficacy of temozolomide (TMZ) in both cellular and animal models [102,125,137]. Reactive oxygen species within GBM and GSCs constitute a critical determinant of therapeutic outcome since ROS-mediated redox signaling and oxidative stress can both drive tumor progression and potentiate sensitivity to therapeutic interventions [138,139] (Table 2).

## 16. Future Directions

The progress achieved after decades of research into the fundamental science of glioblastomas is promptly being converted into novel clinical trials, leveraging improved genetic and epigenetic characterization of GBM, along with a more comprehensive understanding of the brain microenvironment and its interactions with the immune system [139]. These advancements have established the fundamental roles of GSCs and redox signaling in resistance to standard therapies and tumor recurrence. Considering these observations, immunotherapy, encompassing immune checkpoint inhibition, chimeric antigen receptor T-cell (CAR-T) therapy, oncolytic virotherapy, and vaccine-based techniques, has emerged as a viable approach for the treatment of GBM. Combinatorial strategies aim to mitigate immune-related adverse effects while simultaneously augmenting the anticancer immune response [140,141]. In addition, innovative technologies, including focused ultrasound, convection-enhanced delivery, and nanocarrier systems, are being devised to cross the blood–brain barrier (BBB), thereby enabling efficient and targeted therapeutic delivery, particularly in patients with recurrent GBM [142]. Alongside immune-based methods, treatments targeting redox homeostasis in GSCs have attained heightened interest. Modulating ROS levels via glutathione depletion, suppression of redox-buffering systems, or disruption of critical regulators such as Nrf2 and KEAP1 may bolster the sensitivity of GSCs to therapy [143]. Targeting metabolic reprogramming via the EGFR/AKT and mevalonate pathways or modifying cholesterol and fatty acid synthesis, in conjunction with leveraging oxidative stress-induced DNA damage in tumor cells, presents alternative strategies to hinder the progression of GBM [144]. The integration of multi-omics profiling and single-cell transcriptomics has elucidated the heterogeneity of GBM and the plasticity of glioma GSCs, thereby enabling more precise and individualized therapeutic approaches. Future research should focus on the development of redox-based biomarkers for predicting therapeutic responses and the improvement of redox-targeted delivery systems that specifically target GSCs within the adverse tumor microenvironment. However, we found that phase II glioblastoma trials continue to be conducted largely in single-center settings and with single-arm designs, placing the field at risk for continued late-phase trial failures and beleaguered drug development [144]. Existing drug delivery systems based on solid nanoparticles continue to pose a major challenge for glioblastoma chemotherapy. Structural droplet drugs (ESDDs) designed from extracellular vesicles (EVs) have been shown to significantly improve antitumor efficacy against glioblastoma, allowing them to cross the BBB/BBTB highly efficiently [144,145]. An obvious consequence of radiotherapy is that severe and prolonged lymphopenia frequently occurs in patients with glioblastoma after standard chemoradiotherapy and has been associated with poorer survival [146].

## 17. Conclusions

Oxidative stress is pivotal in the progression of glioblastoma, significantly affecting its therapeutic response and overall prognosis. The intricate network of pathways influenced by ROS, including mitochondrial dysfunction, redox-sensitive signaling, glutathione metabolism, and immunological regulation, highlights the dual role of ROS as both promoters of tumor growth and potential therapeutic targets. Glioma stem-like cells, known for their resistance to conventional therapies and adaptability to redox fluctuations, present a significant challenge in GBM treatment. Targeting oxidative stress through glutathione depletion, proteasomal modulation, or EGFR/AKT pathway inhibition offers promising strategies to enhance GSC sensitivity to both established and novel therapies. Further investigation into redox regulation is essential to identify potential pitfalls of emerging treatments. Combining redox biology insights with advanced delivery systems and immunomodulatory approaches may improve therapeutic efficacy and durability.

## Figures and Tables

**Figure 1 brainsci-15-00884-f001:**
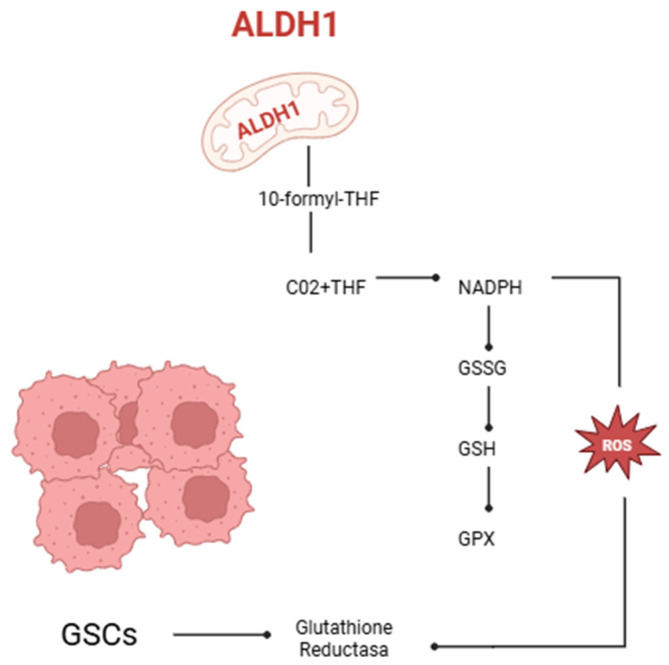
Mitochondrial enzyme aldehyde dehydrogenase 1 family member 2 (ALDH1L2) catalyzes the conversion of 10-formyl-tetrahydrofolate (10-formyl-THF) into carbon dioxide (CO_2_) and tetrahydrofolate (THF), generating reduced nicotinamide adenine dinucleotide phosphate (NADPH). NADPH serves as a cofactor for the enzyme glutathione reductase (GSH reductase), which reduces oxidized glutathione (GSSG) to its active form, reduced glutathione (GSH). Glutathione peroxidase (GPX) then utilizes GSH to convert peroxides into water or alcohol, thereby contributing to cellular redox homeostasis (created with https://BioRender.com). The ALDH1L2 enzyme is crucial for redox balance in glioma stem-like cells (GSCs). By synthesizing NADPH, ALDH1L2 enhances the glutathione antioxidant system, which allows GSCs to protect themselves from oxidative stress and survive in the tumor microenvironment. ALDH1L2 dysfunction compromises antioxidant defenses, increasing intracellular ROS levels and reducing the aggressive phenotype of GSCs. Therefore, ALDH activity not only preserves redox balance and mitochondrial function but also serves as a biomarker for tumor aggressiveness and therapy resistance. This mechanism highlights the importance of redox signaling pathways in GSC maintenance and aggressiveness.

**Figure 2 brainsci-15-00884-f002:**
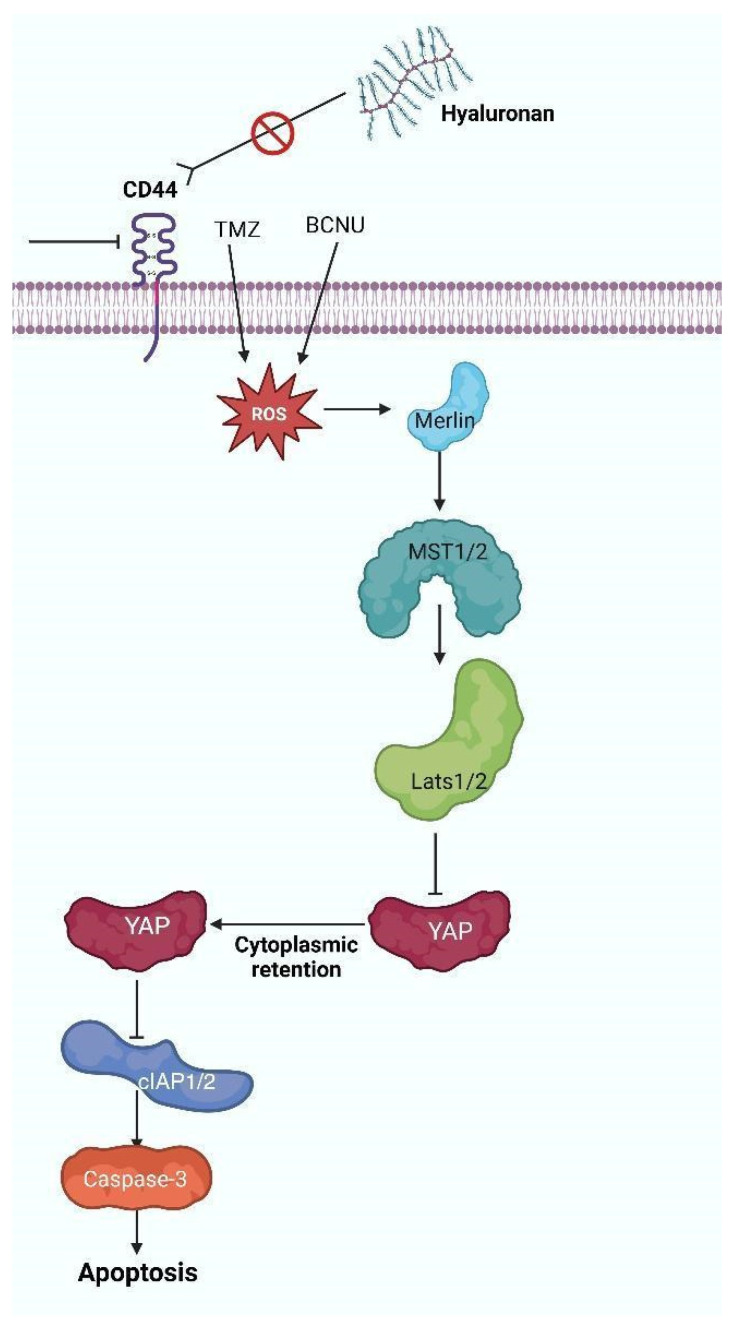
CD44 depletion in glioblastoma cells enhances their response to oxidative stress induced by antitumor agents such as temozolomide (TMZ) and carmustine (BCNU). Reactive oxygen species (ROS) activate the Hippo pathway by phosphorylating the merlin protein, which, in turn, triggers the phosphorylation and activation of Mammalian Sterile Twenty-like kinases 1 and 2 (MST1/2) and Large Tumor Suppressors 1 and 2 (LATS1/2). This cascade leads to the phosphorylation and inactivation of Yes-associated protein (YAP), preventing its nuclear translocation. As a result, the expression of anti-apoptotic proteins cIAP1/2 is inhibited, while levels of cleaved caspase-3 increase, ultimately promoting apoptosis in tumor cells (created with https://BioRender.com).

**Figure 3 brainsci-15-00884-f003:**
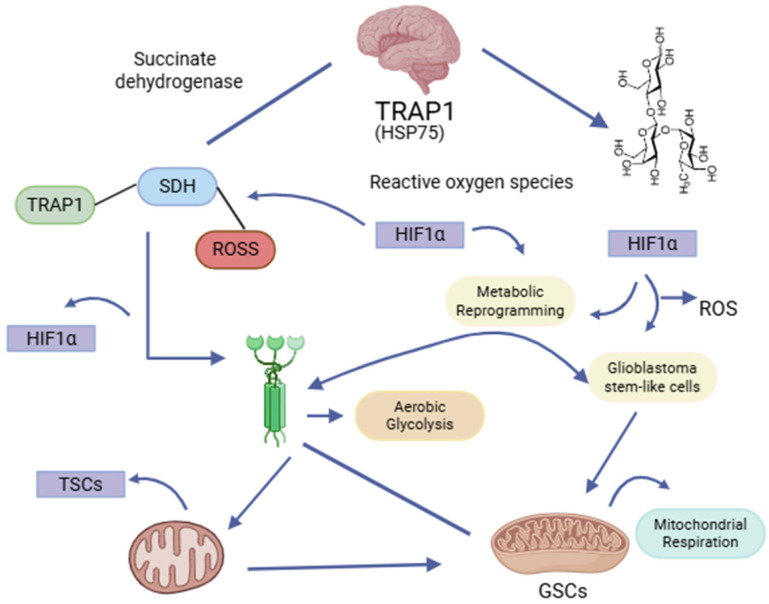
FTRAP1 (HSP75) modulates mitochondrial metabolism by suppressing succinate dehydrogenase (SDH), diminishing the production of reactive oxygen species (ROS), and stabilizing HIF1α. This facilitates metabolic reprogramming towards aerobic glycolysis, enhancing the activation of glioma stem cells (GSCs). Nonetheless, TRAP1 can activate SDH in specific situations, enhancing mitochondrial respiration in GSCs, hence contributing to their metabolic flexibility and treatment resistance (created with https://BioRender.com).

**Figure 4 brainsci-15-00884-f004:**
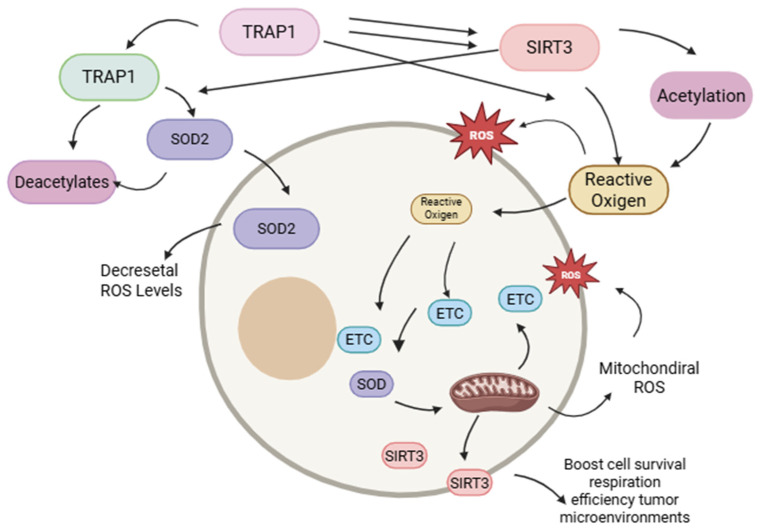
The figure illustrates the role of Sirtuin 3 (SIRT3) as a mitochondrial regulator of redox equilibrium and cellular oxidative stress. Its function in activating antioxidant enzymes such as MnSOD and IDH, modulating ROS, boosts cell survival, respiration, and tumor microenvironment efficiency (created with https://BioRender.com).

**Figure 5 brainsci-15-00884-f005:**
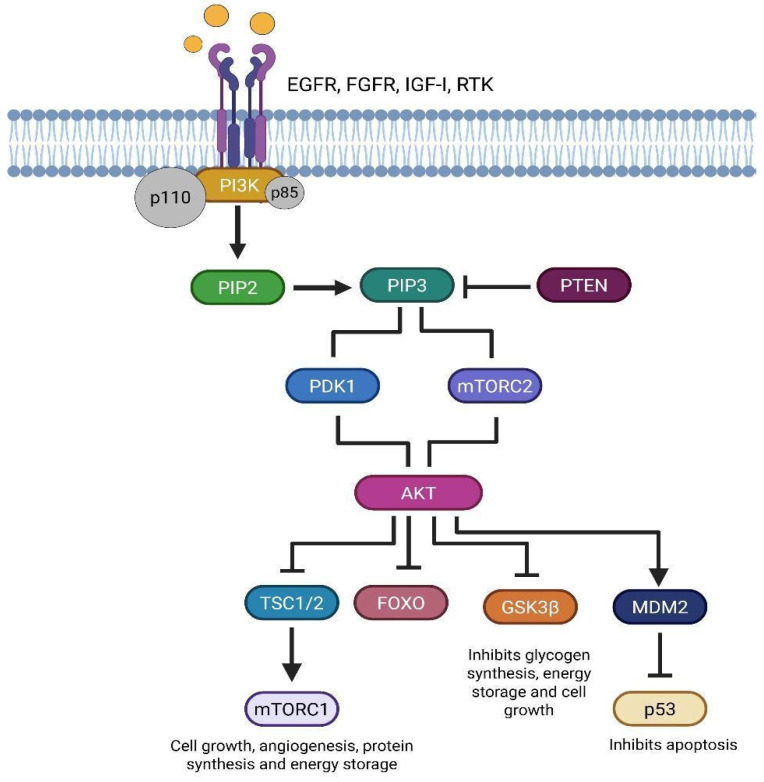
Schematic representation of the bidirectional TRAP1-SIRT3 functional axis in glioma stem cells (GSCs). TRAP1 augments the enzymatic function of SIRT3, facilitating the deacetylation and activation of SOD2, hence diminishing mitochondrial reactive oxygen species (ROS) levels. SIRT3 modulates the acetylation of TRAP1, creating a positive feedback loop between the two proteins. This functional axis enhances mitochondrial respiration efficiency by modulating electron transport (ETC), particularly in low glucose availability situations. The interplay between TRAP1 and SIRT3 enhances GSC metabolic adaptability under stress, hence augmenting their survival, self-renewal ability, and respiratory efficiency in challenging tumor microenvironments (created with https://BioRender.com).

**Figure 6 brainsci-15-00884-f006:**
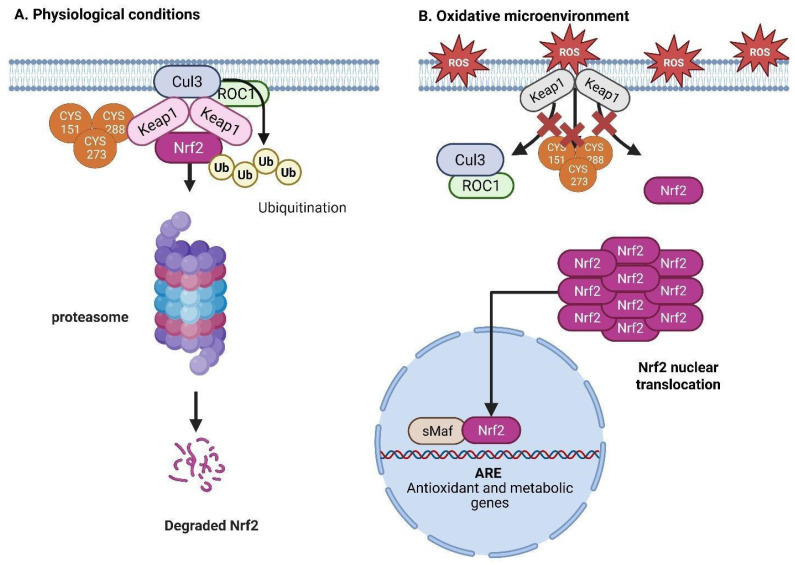
Regulation of Nrf2 under physiological and oxidative stress conditions. (**A**) Under normal physiological conditions, Keap1 binds Nrf2 through cysteine residues (Cys151, Cys273, and Cys288), facilitating its ubiquitination via the Cul3-ROC1 E3 ligase complex, leading to proteasomal degradation. (**B**) In an oxidative microenvironment, reactive oxygen species (ROS) modify the cysteine residues on Keap1, inhibiting its ability to promote Nrf2 ubiquitination. As a result, stabilized Nrf2 accumulates and translocates into the nucleus, where it forms a heterodimer with sMaf proteins and binds to antioxidant response elements (AREs), activating the transcription of antioxidant and metabolic genes (created with https://BioRender.com).

**Figure 7 brainsci-15-00884-f007:**
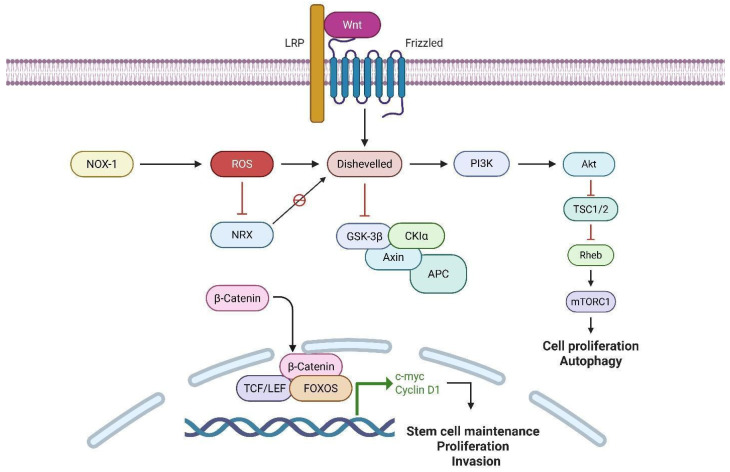
Interaction between Wnt/β-catenin signaling and reactive oxygen species (ROS) in the regulation of cellular functions. Activation of Wnt ligands binds Frizzled and LRP receptors, leading to Dishevelled activation and subsequent inhibition of the β-catenin destruction complex (GSK-3β, Axin, CKIα, and APC). Stabilized β-catenin translocates into the nucleus, where it interacts with TCF/LEF and FOXO transcription factors to promote the expression of target genes such as c-myc and Cyclin D1, supporting stem cell maintenance, proliferation, and invasion. Concurrently, Wnt signaling activates PI3K-Akt, which inhibits TSC1/2, enabling Rheb-mediated activation of mTORC1, thereby promoting cell proliferation and autophagy. ROS, produced by NOX-1, enhance Wnt signaling by inhibiting NRX, a negative regulator of Dishevelled, further amplifying β-catenin signaling (created with https://BioRender.com).

**Figure 8 brainsci-15-00884-f008:**
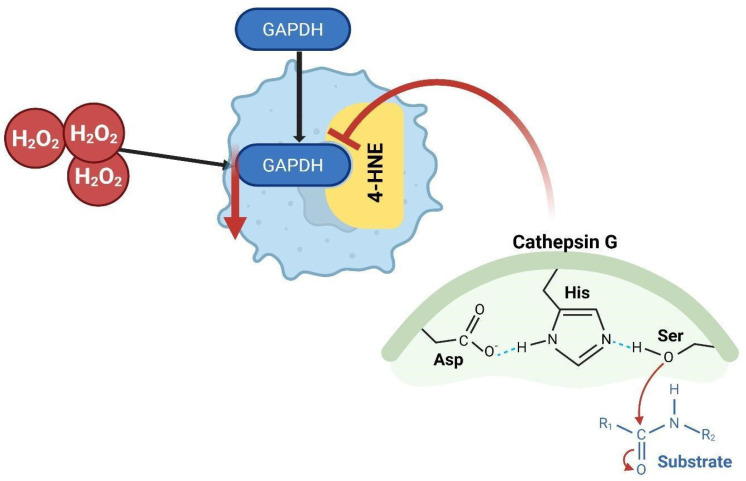
Degradation of oxidatively modified GAPDH under oxidative stress. Hydrogen peroxide (H_2_O_2_) exposure leads to oxidative modification of glyceraldehyde-3-phosphate dehydrogenase (GAPDH), contributing to decreased protein levels and enzymatic activity. Likewise, a major modification is by 4-hydroxy-2-nonenal (4-HNE), a cytotoxic lipid peroxidation byproduct that covalently binds to GAPDH. Cathepsin G, a serine protease featuring a catalytic triad (Asp-His-Ser), is implicated in the degradation of 4-HNE-modified GAPDH (created with https://BioRender.com).

**Figure 9 brainsci-15-00884-f009:**
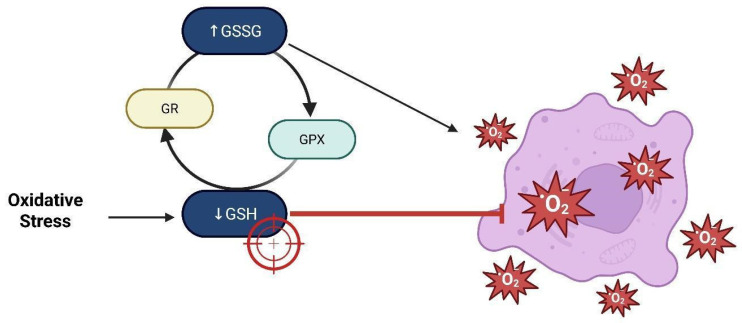
Glutathione redox cycling in the oxidative tumor microenvironment. Elevated oxidative stress in the tumor microenvironment (TME) leads to depletion of reduced glutathione (GSH), resulting in accumulation of its oxidized form (GSSG). Glutathione peroxidase (GPX) catalyzes the reduction of reactive oxygen species (ROS) using GSH, while glutathione reductase (GR) regenerates GSH from GSSG to maintain redox homeostasis. Excessive ROS, if not neutralized, causes extensive damage to DNA, proteins, and lipids, promoting cellular dysfunction and death. Targeting the GSH system has emerged as a promising redox-sensitive therapeutic approach, especially in tumors with elevated GSH levels that confer resistance to oxidative damage (created with https://BioRender.com).

**Table 1 brainsci-15-00884-t001:** Functional comparison of RIOK1 and RIOK2 in glioblastoma and their relationship to the AKT/c-Myc pathway. Summary of the main functions of RIOK1 and RIOK2 in ribosomal maturation, their interaction with the AKT pathway, and the effects associated with their overexpression and inhibition in glioblastoma cells.

	Main Function	Interaction with AKT	Effects in Glioblastoma	Consequences of Inhibition
RIOK1	Participates in the final maturation of the 40S ribosomal subunit in the cytoplasm.	Its expression is induced by AKT; correlates with AKT activation.	Overexpressed in high-grade gliomas; associated with increased migration and invasiveness.	Decreased levels of AKT1 and c-Myc; reduced tumor cell migration and invasiveness.
RIOK2	Facilitates nuclear export of the pre-40S subunit and its maturation in the cytoplasm (via ATPase activity after phosphorylation).	Positive feedback loop: RIOK2 activates AKT and vice versa.	Overexpressed in glioblastoma cells; contributes to tumor progression.	Indirect reduction of AKT activation and proliferative signals.
c-Myc	Regulates pericellular adhesion and genes associated with invasiveness and metastasis.	Activated by AKT.	Associated with tumor progression and aggressiveness in glioblastoma.	Decreased expression following RIOK1 inhibition.

**Table 2 brainsci-15-00884-t002:** Therapeutic strategy in glioblastoma.

Therapeutic Strategy	Main Mechanism of Action	Specific Focus/Objective	Implication in Glioblastoma
Degradation of oxidized proteins	Removal of damaged proteins like 4-HNE-GAPDH	Proteasome, cathepsin G	Crucial for removing proteins damaged by oxidative stress
Targeting NR4A2 in the tumor immune microenvironment	Modulation of microglial plasticity; reduction of tumor proliferation	NR4A2, SQLE, c-Myc	Enhances antigen-presenting capability of microglia; reduces tumor proliferation
Glutathione (GSH) depletion	Interference with the GSH antioxidant system	Use of nanoparticles with disulfide bonds	Increases chemotherapy sensitivity in GSCs
Inhibition of the EGFR/AKT pathway	Disruption of energy metabolism	EGFR, EGFRvIII, MK-2206, MK-803, TCA cycle, ATP synthesis	Decreases tumor growth; enhances temozolomide efficacy

The table shows various therapeutic strategies aimed at glioblastoma stem cells (GSCs) to combat treatment resistance, focusing on the modulation of oxidative stress. Approaches such as the degradation of oxidized proteins, intervention in the tumor immune microenvironment (via NR4A2), glutathione depletion, and inhibition of the EGFR/AKT pathway are detailed.

## Data Availability

Not applicable.

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
