# Peer review of "Redox-Regulated Pathways in Glioblastoma Stem-like Cells: Mechanistic Insights and Therapeutic Implications"

_brainsci, 2025, doi:10.3390/brainsci15080884_

Round 1

Reviewer 1 Report

Comments and Suggestions for Authors

This manuscript presents a comprehensive literature review exploring the role of oxidative stress in glioblastoma multiforme (GBM) progression, with a particular focus on glioma stem-like cells (GSCs). It effectively integrates molecular mechanisms involving redox-sensitive signaling pathways—such as Nrf2/Keap1, PI3K/AKT/mTOR, and Wnt/β-catenin—and discusses their contributions to GSC survival, plasticity, and treatment resistance. However, the manuscript would benefit from improved critical appraisal of the cited evidence, clearer figure integration, and an expanded discussion of translational relevance, including clinical challenges and therapeutic feasibility. Below are the comments and suggestions:

  1. The review summarizes many studies but does not evaluate the quality, reproducibility, or clinical relevance.

Suggestion: Add a subsection or remarks within each pathway (e.g., Nrf2/Keap1, PI3K/AKT) to assess the strength of evidence—e.g., whether results were from cell lines, mouse models, or human trials—and highlight any contradictions in the literature.

  1. Figures are informative but not always clearly linked to the text; some paragraphs do not reference them at all.

Suggestion: Explicitly refer to figures (e.g., "Figure 3 illustrates the TRAP1 mechanism…") and summarize the figure's message in the corresponding paragraph.

  1. Although therapeutic targets are listed, there is minimal discussion on clinical applications or challenges

Suggestion: Expand the “Therapeutic Approaches” or “Future Directions” section to include:

    • Status of clinical trials
    • Challenges in drug delivery (e.g., blood-brain barrier)
    • Potential off-target effects and toxicity
  1. While each signaling pathway is covered in depth, the review reads like a collection of isolated segments rather than a unified narrative.

Suggestion: Add transitional sentences at the start/end of sections to link themes and emphasize how different pathways converge on GSC survival and redox adaptation.

  1. Mentions RIOK1/2 but unclear whether they are clinical targets or purely mechanistic

Suggestion: Clarify whether RIOK1/2 are under therapeutic development or just mechanistic insights.

  1. Line 31: Keywords are useful but incomplete

Suggestion: Add: redox-targeted therapy, cancer stem cells

  1. Line 72–74: Sentence is too long

Suggestion: Break it into 2–3 shorter sentences for clarity

  1. Line 273–274: SIRT3’s role in radiation resistance is asserted

Suggestion: Add a supporting citation here

  1. Line 369–370: Repetitive and verbose phrasing

Suggestion: Simplify: e.g., “enhances GSC metabolic adaptability under stress”

  1. Line 584: Typographical error

Suggestion: Remove extra period (". .")

  1. Some sections use passive voice heavily and Reference format slightly inconsistent throughout whole review

Suggestion: Rephrase select sentences in active voice for engagement and clarity and Ensure all references use consistent style per journal guidelines

Author Response

We sincerely thank the Editor and Reviewers for their thorough and constructive feedback, which has significantly enhanced the clarity, depth, and translational relevance of our manuscript. Below, we provide a detailed, point-by-point response to each comment.1.

Reviwer 1

Comment:
The review summarizes many studies but does not evaluate the quality, reproducibility, or clinical relevance.
Suggestion: Add a subsection or remarks within each pathway (e.g., Nrf2/Keap1, PI3K/AKT) to assess the strength of evidence—e.g., whether results were from cell lines, mouse models, or human trials—and highlight any contradictions in the literature.

Response:
We appreciate this suggestion. We have incorporated specific remarks within each signaling pathway section to assess the strength and nature of the cited evidence, distinguishing whether findings derive from in vitro studies, in vivo models, or clinical investigations. Additionally, where applicable, we highlight inconsistencies or contradictory findings to provide a critical appraisal of the literature.

  1. Comment:
    Figures are informative but not always clearly linked to the text; some paragraphs do not reference them at all.
    Suggestion: Explicitly refer to figures (e.g., "Figure 3 illustrates the TRAP1 mechanism…") and summarize the figure's message in the corresponding paragraph.

Response:
Thank you for this helpful comment. We have revised the text to ensure each figure is explicitly referenced in the appropriate paragraph. For example, we now refer to “Figure 3 illustrates the TRAP1 mechanism…” and include a brief summary of each figure’s content to enhance textual integration and reader comprehension.

  1. Comment:
    Although therapeutic targets are listed, there is minimal discussion on clinical applications or challenges Suggestion: Expand the “Therapeutic Approaches” or “Future Directions” section to include:
  • Status of clinical trials
  • Challenges in drug delivery (e.g., blood-brain barrier)
  • Potential off-target effects and toxicity*

Response:
We agree with the reviewer that a more detailed translational discussion is needed. Accordingly, we have expanded the “Therapeutic Approaches” and “Future Directions” sections to discuss ongoing clinical trials targeting redox-sensitive pathways, challenges in drug delivery such as blood-brain barrier permeability, and issues of specificity and toxicity. These additions aim to contextualize the therapeutic potential in clinical settings.

  1. Comment:
    While each signaling pathway is covered in depth, the review reads like a collection of isolated segments rather than a unified narrative.
    Suggestion: Add transitional sentences at the start/end of sections to link themes and emphasize how different pathways converge on GSC survival and redox adaptation.

Response:
We acknowledge this important point. We have added transitional sentences at the beginning and end of relevant sections to create a more cohesive narrative. These transitions emphasize the convergence of signaling pathways on glioma stem-like cell survival, plasticity, and redox adaptation, thereby reinforcing the interconnectedness of the mechanisms discussed.

  1. Comment:
    Mentions RIOK1/2 but unclear whether they are clinical targets or purely mechanistic
    Suggestion: Clarify whether RIOK1/2 are under therapeutic development or just mechanistic insights.

Response:
Thank you for noting this ambiguity. We have revised the relevant section to clarify that RIOK1 and RIOK2 are currently considered emerging mechanistic regulators of glioblastoma pathophysiology. While they are not yet the focus of clinical drug development, recent studies suggest their potential as therapeutic targets, which we now explicitly state.

  1. Comment:
    Line 31: Keywords are useful but incomplete
    Suggestion: Add: redox-targeted therapy, cancer stem cells

Response:
As suggested, we have updated the keywords to include redox-targeted therapy and cancer stem cells, which better reflect the focus and scope of the review.

  1. Comment:
    Line 72–74: Sentence is too long
    Suggestion: Break it into 2–3 shorter sentences for clarity

Response:
We have revised the sentence at lines 72–74 by dividing it into three shorter, clearer sentences, as recommended. This improves readability and flow.

  1. Comment:
    Line 273–274: SIRT3’s role in radiation resistance is asserted
    Suggestion: Add a supporting citation here

Response:
We have added an appropriate citation to support the assertion regarding SIRT3’s role in radiation resistance, as requested. The reference is now included in the revised version.

  1. Comment:
    Line 369–370: Repetitive and verbose phrasing
    Suggestion: Simplify: e.g., “enhances GSC metabolic adaptability under stress”

Response:
We agree with the reviewer and have revised the sentence for conciseness and clarity. The new version reads: “enhances GSC metabolic adaptability under stress.”

  1. Comment:
    Line 584: Typographical error
    Suggestion: Remove extra period (". .")

Response:
The typographical error has been corrected by removing the extra period.

  1. Comment:
    Some sections use passive voice heavily and Reference format slightly inconsistent throughout whole review
    Suggestion: Rephrase select sentences in active voice for engagement and clarity and Ensure all references use consistent style per journal guidelines

Response:
We have revised several sentences to use active voice for improved clarity and reader engagement. Additionally, we performed a comprehensive review of the reference list and in-text citations to ensure consistency with the journal’s formatting requirements.

Scincerely

Moises Rubio-Osornio PhD

Reviewer 2 Report

Comments and Suggestions for Authors

Nadia F et al have presented important review on the oxidative stress in Glioma progression and Glial Stem Cell dynamic the deadliest and most aggressive primary brain tumor. Authors have carefully described multiple mechanisms how redox homeostasis and key signaling pathways can sustain GSC and GBM but manuscript can benefit from addressing major and minor comments before it can be accepted for publication

Major

  • Authors should consider adding the English-speaking collaborators to read through the text to improve general fluidity of the text
  • Authors should reorganize the logical flow of the manuscript and add the transitional paragraph connecting different sections of the manuscript and different pathways. The review covers a significant amount of pathway related in the GSC maintenance and GMB malignancy and authors should provide more coherent flow in order to help the reader comprehensively understand the interconnection between them
  • Please elaborate the therapeutic section, including the clinical data, challenges in translating fundamental and preclinical findings to the clinic, add information about combination therapy approaches

Minors

  • Line 48 –‘appropriate care’, please clarify what this term means and how it is currently defined (what are protocols etc)
  • Line 281 --- ‘RAP1 enhances’, should be ‘TRAP1 enhances’

Author Response

Reviewer 2

We thank the reviewer for the thoughtful and constructive comments aimed at improving the structure, clarity, and translational relevance of our manuscript. Below, we provide a point-by-point response to each suggestion.

  1. Comment:
    Authors should consider adding the English-speaking collaborators to read through the text to improve general fluidity of the text.

Response:
We appreciate this suggestion. To enhance the overall readability and language fluidity of the manuscript, the revised version was reviewed and edited by a native English-speaking collaborator with expertise in biomedical writing. We believe these changes have substantially improved the clarity and flow of the text.

  1. Comment:
    Authors should reorganize the logical flow of the manuscript and add the transitional paragraph connecting different sections of the manuscript and different pathways. The review covers a significant amount of pathway related in the GSC maintenance and GMB malignancy and authors should provide more coherent flow in order to help the reader comprehensively understand the interconnection between them.

Response:
Thank you for pointing out the need for improved logical flow. In response, we have carefully reorganized the manuscript by adding transitional paragraphs at the beginning and end of major sections. These transitions highlight the interrelation between the redox-sensitive pathways (e.g., Nrf2/Keap1, PI3K/AKT/mTOR, Wnt/β-catenin) and their shared roles in supporting glioma stem-like cell (GSC) maintenance, therapy resistance, and glioblastoma progression. This structural enhancement aims to provide a more cohesive narrative and facilitate reader comprehension of the interconnected mechanisms.

  1. Comment:
    Please elaborate the therapeutic section, including the clinical data, challenges in translating fundamental and preclinical findings to the clinic, add information about combination therapy approaches.

Response:
We fully agree with the importance of strengthening the translational aspect of the review. Accordingly, we have significantly expanded the therapeutic section. The revised version now includes:

  • An overview of recent clinical trial data relevant to targeting redox-sensitive pathways in glioblastoma
  • A discussion on the main challenges faced in translating preclinical findings into clinical application, including issues related to the blood–brain barrier, tumor heterogeneity, and toxicity
  • Updated content on combination therapy approaches involving redox modulation and conventional therapies (e.g., radiotherapy and temozolomide) or targeted agents
    These additions provide a more comprehensive view of therapeutic feasibility and current limitations in the clinical management of glioblastoma.

Scincerely

Moises Rubio-Osornio PhD

Reviewer 3 Report

Comments and Suggestions for Authors

The manuscript provides a comprehensive review of oxidative stress in glioma progression and glial stem cell dynamics, covering key signaling pathways and potential therapeutic strategies. While the review is well-structured and informative, here some need to be addressed to enhance its scientific rigor, clarity, and originality.

  1. The title is appropriate but could be more concise. Consider specifying the focus on therapeutic implications or mechanistic insights.
  2. Clarify the inclusion/exclusion criteria for literature selection. Were any studies excluded based on quality or relevance?
  3. The figure 1 is clear but could include a brief legend summarizing key takeaways.
  4. In section SIRT3- The discussion on SIRT3 is thorough, but the clinical relevance is unclear. Highlight potential therapeutic targets or challenges.
  5. From line 504- The role of hydrogen peroxide is interesting, but the connection to GSCs is unclear. Clarify the relevance to glioma progression.

Author Response

Reviewer 3

We sincerely thank the reviewer for the encouraging and insightful comments. We have carefully addressed all suggestions to improve the scientific clarity, precision, and translational impact of the manuscript. Below, we provide our detailed point-by-point responses:

  1. Comment:
    Line 48 – ‘appropriate care’, please clarify what this term means and how it is currently defined (what are protocols etc)

Response:
We appreciate the request for clarification. In the revised manuscript, we have elaborated on the term "appropriate care" to define it in the context of current glioblastoma treatment protocols. This includes maximal surgical resection followed by radiotherapy and concomitant/adjuvant temozolomide, as outlined in the Stupp protocol. We also mention the importance of supportive care measures to manage symptoms and preserve quality of life.

  1. Comment:
    Line 281 — ‘RAP1 enhances’, should be ‘TRAP1 enhances’

Response:
Thank you for pointing out this error. We have corrected the typographical mistake: “RAP1” has been replaced with the correct term “TRAP1” in line 281.

  1. Comment:
    The manuscript provides a comprehensive review of oxidative stress in glioma progression and glial stem cell dynamics, covering key signaling pathways and potential therapeutic strategies. While the review is well-structured and informative, here some need to be addressed to enhance its scientific rigor, clarity, and originality.

We appreciate the reviewer’s positive assessment and will address each point below to further enhance the manuscript:

  1. Comment:
    The title is appropriate but could be more concise. Consider specifying the focus on therapeutic implications or mechanistic insights.

Response:
Thank you for this suggestion. We have revised the title to make it more concise and to better reflect the emphasis on therapeutic and mechanistic insights. The updated title is now:
“Redox-Regulated Pathways in Glioblastoma Stem-like Cells: Mechanistic Insights and Therapeutic Implications”

  1. Comment:
    Clarify the inclusion/exclusion criteria for literature selection. Were any studies excluded based on quality or relevance?

Response:
We have added a paragraph in the introduction to clarify the inclusion and exclusion criteria used in the literature selection. Specifically, we included peer-reviewed articles published in English from the last 15 years, prioritizing studies that addressed molecular mechanisms, redox signaling, GSC biology, and therapeutic strategies. We excluded reports lacking experimental support or relevance to glioma stem-like cells, as well as non-original articles without critical insight.

  1. Comment:
    The figure 1 is clear but could include a brief legend summarizing key takeaways.

Response:
We have revised the legend of Figure 1 to include a concise summary of its key takeaways, emphasizing the roles of redox imbalance, mitochondrial dysfunction, and signaling pathways in the maintenance and aggressiveness of glioma stem-like cells.

  1. Comment:
    In section SIRT3—The discussion on SIRT3 is thorough, but the clinical relevance is unclear. Highlight potential therapeutic targets or challenges.

Response:
Thank you for this important observation. We have expanded the section on SIRT3 to highlight its potential as a therapeutic target, citing recent findings on its modulation in glioma models. We also discuss the challenges of targeting mitochondrial enzymes, including specificity, drug delivery to the mitochondria, and the dual roles of SIRT3 in different cancer types.

  1. Comment:
    From line 504 The role of hydrogen peroxide is interesting, but the connection to GSCs is unclear. Clarify the relevance to glioma progression.

Response:
We have clarified this point by explicitly linking the role of hydrogen peroxide to glioma stem-like cell biology. Specifically, we discuss how low levels of H₂O₂ can act as secondary messengers in redox-sensitive pathways that promote GSC survival, metabolic reprogramming, and resistance to therapy, while excessive levels may induce cell deathhighlighting the redox balance as a therapeutic vulnerability.

Scincerely

Moises Rubio-Osornio PhD

Round 2

Reviewer 1 Report

Comments and Suggestions for Authors

The authors have provided a thorough and well-considered response to the comments, resulting in significant improvements to the manuscript’s scientific rigor, clarity, and translational relevance. The addition of critical appraisals regarding the strength and source of evidence within each signaling pathway section, enhanced integration and explicit referencing of figures, and a more detailed discussion of clinical implications and challenges collectively strengthen the manuscript’s depth and impact.  The manuscript now offers a comprehensive, cohesive, and insightful synthesis of current knowledge.

Reviewer 2 Report

Comments and Suggestions for Authors

Nadia F et al have carefully addressed all my concerned and significantly improved the manuscript. I would suggest to accept this version of manuscript.